# Abdominal Organ Segmentation via Self Training

Hexin Dong[1], Zifan Chen[1], Jie Zhao[3], Mingze Yuan[1], Fei Yu[1], Jiangdong Zhang[1], and Li Zhang[1,2]

[1] Center for Data Science, Peking University, Beijing, China
[2] Center for Data Science in Health and Medicine, Peking University, Beijing, China
[3] National Engineering Laboratory for Big Data Analysis and Applications, Peking University, Beijing, China `{donghexin,zhangli_pku}@pku.edu.cn`

**Abstract.** This paper proposes a semi-supervised abdominal organ segmentation approach based on self-training. Self-training(ST) helps improve the segmentation network's decision boundary with unlabeled images and generated pseudo labels. Experiment results show that our method has significantly outperformed the non-ST baseline, improving the mean Dice score from 0.8195 to 0.8568.

**Keywords:** Semantic Segmentation · Self Training

## 1 Introduction

Abdomen organ segmentation has many important clinical applications, such as organ quantification, surgical planning, and disease diagnosis. However, manually annotating organs from CT scans is time-consuming and labor-intensive. Thus, we usually cannot obtain a huge number of labeled cases. As a potential alternative, semi-supervised learning can explore useful information from unlabeled cases.

In this challenge, we propose an effective semi-supervised semantic segmentation method via self-training. Self-training or self-distillation has shown impressive results in recent years [9,1,10]. In the field of semi-supervised semantic segmentation, CNN-based self-training methods mainly fine-tune a trained segmentation model using the unlabeled images and the pseudo labels, which improves the model's decision boundary.

## 2 Method

### 2.1 Preprocessing

– First, we observe that there are redundant slices (including legs, feet, shoulder, etc.) in some cases. We detect those slices via the Connected Component algorithm and cut them out.
– Limited by memory, we cut each case into several parts (48 slices per case).
– We do other preprocessing following nnUNet standard preprocessing stage.

## 2.2    Proposed Method

For 3D segmentation, we follow the nnUNet framework  [5]. Several research settings are implemented.Figure 1 illustrates the applied 3D nnU-Net [5], where a U-Net architecture is adopted.

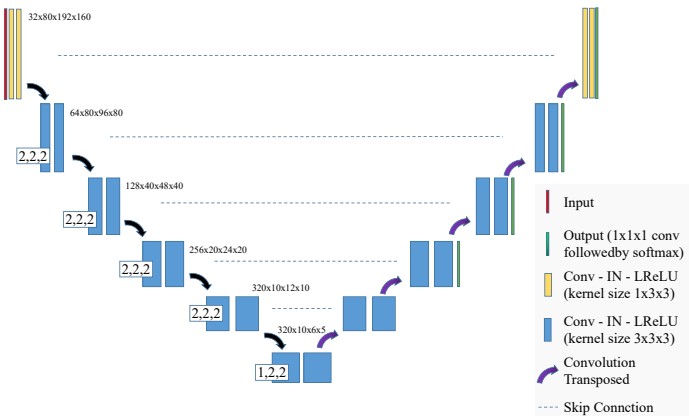

**Fig. 1.** Network architecture

Second, we further apply self-training to improve the decision boundary of the segmentation model. Similar to [6], we introduce a super parameter $q$ of the pixel portion. We iteratively generate the pseudo label $\hat{y}_c$ using the top $q$ of pixels in segmentation output $y_c$ with a higher probability of retraining the model. The overall training process of the proposed method is summarized in Algorithm 1.

---

**Algorithm 1** training process of the proposed method

---

1:  Initialize labeled scans images and label $(X_s, y_s)$, unlabeled images $X_t$, Segmentation network $S$

2:  Train network $S$ with $(X_s, y_s)$

3:  Initialize concat scans images $X_c = \{X_s, X_t\}$, self-training segmentation network $S_0 = S$

4:  **for** $k \leftarrow 1$ to $K$ **do**

5:      input $X_c$ into $S_{k-1}$ and generate pseudo label $\hat{y_c^k}$ with a fixed portion $q_k$

6:      Initialize $S_k \leftarrow S_{k-1}$

7:      Train $S_k$ with $(X_c, \hat{y_c^k})$

8:  **end for**

9:  **return** $S_k$

---

Loss function: Following nnUNet, we use the summation between Dice loss and cross-entropy loss.

### 2.3  Post-processing

We concatenate the grouped slices together as the final output.

## 3  Experiments

### 3.1  Dataset and evaluation measures

The FLARE2022 dataset is curated from more than 20 medical groups under the license permission, including MSD [8], KiTS [3,4], AbdomenCT-1K [7], and TCIA [2]. The training set includes 50 labeled CT scans with pancreas disease and 2000 unlabelled CT scans with liver, kidney, spleen, or pancreas diseases. The validation set includes 50 CT scans with liver, kidney, spleen, or pancreas diseases. The testing set includes 200 CT scans where 100 cases have liver, kidney, spleen, or pancreas diseases and the other 100 cases have uterine corpus endometrial, urothelial bladder, stomach, sarcomas, or ovarian diseases. All the CT scans only have image information, and the center information is not available.

The evaluation measures consist of two accuracy measures: Dice Similarity Coefficient (DSC) and Normalized Surface Dice (NSD), and three running efficiency measures: running time, area under GPU memory-time curve, and area under CPU utilization-time curve. All measures will be used to compute the ranking. Moreover, the GPU memory consumption has a 2 GB tolerance.

### 3.2  Implementation details

**Environment settings** The development environments and requirements are presented in Table 1.

**Table 1.** Development environments and requirements.

| | |
|---|---|
| Ubuntu version | CentOS Linux release 7.9.2009 |
| CPU | Intel(R) Xeon(R) CPU E5-2620 v4 @ 2.10GHz |
| RAM | 160GB |
| GPU (number and type) | Four NVIDIA RTX3090 24G |
| CUDA version | 11.3 |
| Programming language | Python 3.8 |
| Deep learning framework | Pytorch (Torch 1.11, torchvision 0.12.0) |

**Table 2.** Training protocols.

| Network initialization | "he" normal initialization |
|---|---|
| Batch size | 2 |
| Patch size | $32 \times 384 \times 384$ |
| Total epochs | 900 |
| Optimizer | SGD with nesterov momentum ($\mu = 0.99$) |
| Initial learning rate (lr) | 0.01 |
| Lr decay schedule | cosine scheduler |
| Training time | 96 |

**Training protocols** Table 2 describes our training protocols. Besides, we set $K = 2$ in self training stage and we only use 200 unlabeled images.

### 3.3   Results

The experiments show that self-training achieves better performance on overall Dice. As shown in Table 3, it improves Mean Dice Score from 0.8195 to 0.8568.

**Table 3.** DSC scores(%) for selected model.

| Model Name | Liver | RK | Spleen | Pancreas | Aorta | IVC | RAG | LAG | Gallbladder | Esophagus | Stomach | Duodenum | LK | Mean DSC |
|---|---|---|---|---|---|---|---|---|---|---|---|---|---|---|
| nnUNet w/o ST | 93.9 | 86.7 | 85.5 | 80.0 | 94.2 | 85.6 | 79.5 | 73.2 | 68.8 | 81.4 | 81.8 | 70.6 | 81.5 | 81.8 |
| nnUNet with ST | 94.5 | 85.7 | 91.2 | 87.0 | 95.9 | 90.0 | 80.6 | 81.2 | 71.8 | 86.8 | 88.1 | 77.2 | 84.1 | 85.7 |

**Acknowledgements** The authors of this paper declare that the segmentation method they implemented for participation in the FLARE 2022 challenge has not used any pre-trained models nor additional datasets other than those provided by the organizers. The proposed solution is fully automatic without any manual intervention.

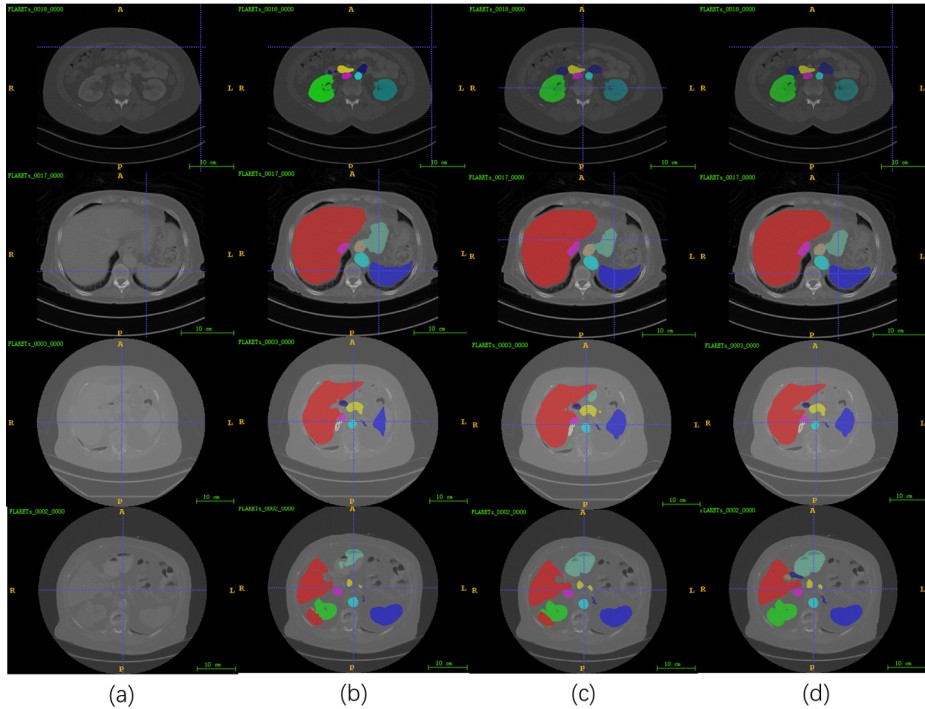

|      |      |      |      |
| (a)  | (b)  | (c)  | (d)  |

**Fig. 2.** Visual results of segmentation output : a).Input Image b).nnUNet output without ST c).nnUNet with ST d).Ground Truth; Sample id : 1).17 2).18 3).2 4).3

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
