# OpenReview forum: "Abdominal Organ Segmentation via Self Training"
_MICCAI.org/2022/Challenge/FLARE_

### Official Review · Reviewer_F1eP · 2022-09-13
**clear description of the method**

**Rating:** 7
**Confidence:** 5

**Review:**

1. The descriptions of the method and the network are clear.
2. The authors need to highlight their contributions and results.

---

### Official Review · Reviewer_HEp7 · 2022-09-15
**Layout issue**

**Rating:** 8
**Confidence:** 3

**Review:**

* In Table 2, the hyper-parameters of lr decay schedule should be declaim.
* In Table 3, the table column names should be tapped.
* In Fig 2, the ids of image is not very clear.

Over all, good job but still need to refine.

---

### Official Review · Reviewer_ktr8 · 2022-09-16
**Abdominal Organ Segmentation via Self Training**

**Rating:** 5
**Confidence:** 4

**Review:**

Strengths: The proposed method achieves effective semi-supervised learning via self-training with a mean DSC of 0.8568.

Weaknesses: The two main aspects of the FLARE22 challenge are efficiency and how to use unlabeled data. Although the proposed method was a very effective semi-supervised learning strategy, the paper does not present any methods or results regarding memory usage or inference time, which would be a bit out of line with the purpose of this challenge. Adding the segmentation efficiency analysis results would be sufficient for acceptance.

---

### Official Review · Reviewer_DGNn · 2022-09-17
**This paper lacks a description of the prediction efficiency. In addition, this paper is too short.**

**Rating:** 4
**Confidence:** 3

**Review:**

In this paper, a semi-supervised method using nnUNet to generate pseudo labels is proposed to continuously improve the segmentation effect of the model. However, this paper does not mention the final prediction part of the network and the prediction efficiency of the network. If the original framework of nnUNet is used as the prediction, it will be very time-consuming. In addition, this paper is too short.

---

### Official Review · Reviewer_v28i · 2022-09-18
**MICCAI-FLARE**

**Rating:** 5
**Confidence:** 4

**Review:**

Weakness:
1. Fig.1 needs to be redrawn, please do not use the example figure in the template.
2. Describe your self-Training method and draw the flow picture in the method section.


Advice:
1. In Fig.2, better adjust CT Hu to [-325,-325].

---

### Official Review · Reviewer_SR7S · 2022-09-19
**Review of Abdominal Organ Segmentation via Self Training**

**Rating:** 7
**Confidence:** 3

**Review:**

The paper is clear enough. Authors prove the efficiency of self-training in FLARE2022.

---

### Official Review · Reviewer_tpJg · 2022-09-19
**This article simply explained their method while the details and more information should be included.**

**Rating:** 6
**Confidence:** 3

**Review:**

Pros:1. This method uses a simple but effective self-training method to improve the semi-supervised performance.
Cons: 1. In the Method section, it’s said that the preprocessing cut the case into slices, however the proposed method illustrate a 3D segmentation framework. Does the slices indicate the sub-part like patches?
2. This training only use 200 unlabeled images, and there is no explanation for the usage of only 200 unlabeled images and how these images were selected.

---

### Meta-Review · Program_Chairs · 2022-09-28

**Recommendation:** Major Revision
**Confidence:** 5

**Metareview:**

Reviewers raise many concerns and suggestions. Please address all comments in the revised manuscript.